

# Can rime splintering explain the ice production in Arctic mixed-phase clouds?

Tomi Raatikainen[1], Silvia Calderón[2], Emma Järvinen[3], Marje Prank[1], and Sami Romakkaniemi[2]

[1]Climate System Research Unit, Finnish Meteorological Institute, Helsinki 00560, Finland
[2]Atmospheric Research Centre of Eastern Finland, Finnish Meteorological Institute, Kuopio 70211, Finland
[3]Institute for Atmospheric and Environmental Research, University of Wuppertal, Wuppertal, Germany

**Correspondence:** Tomi Raatikainen (tomi.raatikainen@fmi.fi)

**Abstract.** Secondary ice production (SIP) can increase ice crystal number concentration (ICNC) by several orders of magnitude, particularly in clean clouds with low concentrations of ice-nucleating particles (INPs). The most common SIP process in models is rime splintering (RS) also called as the Hallett-Mossop process. The generally adopted RS-formulation gives 350 splinters per milligram of rimed ice at the temperature of 268 K. We used large-eddy simulations to examine if rime splintering could explain the high ICNC observed during the ACLOUD (Arctic CLoud Observations Using airborne measurements during polar Day) campaign where cloud temperatures close to 268 K are favourable for rime splintering. With the default model setup, the splinter production rate had to be multiplied by a factor ten to close the gap between modelled and observed ICNCs. Similar changes have been made in other modelling studies. The factor of ten multiplier helped to trigger SIP so that it became a self-sustaining process, fully independent of the primary freezing initiated by INPs. Our simulations reached realistic steady-state ICNCs and maintained stable mixed-phase clouds through the 24-hour simulation time. Additional sensitivity tests showed that the efficiency of SIP depends strongly on model parametrizations and air temperature, so that simulations with a modified setup were able to reproduce the observed ICNCs without the factor of ten multiplier.

## 1 Introduction

Shallow mixed-phase clouds (MPCs) are common over cooler marine regions (Mioche et al., 2015; Listowski et al., 2019). Their important role in the formation of precipitation and the radiation budget make them highly sensitive elements in global climate and weather models (McFarquhar and Cober, 2004; Prenni et al., 2007; Cesana and Storelvmo, 2017; Korolev et al., 2017). Clouds reflect most of the incoming short-wave solar radiation, but they also cause long-wave warming, which can cause sea-ice melting over high-latitudes. This is one of the main drivers of the Arctic amplification (Previdi et al., 2021).

Global climate models struggle representing MPCs mainly because of low spatial and temporal resolution and simplified cloud microphysics. MPC are inherently unstable thermodynamic systems, highly sensitive to turbulent surface fluxes and aerosol perturbations in the cloud condensation nuclei (CCN) and ice-nucleating particles (INP) number concentrations (Eirund et al., 2019; Gierens et al., 2020). Another challenge in modelling MPC comes from their susceptibility to experience secondary ice production (SIP), which can produce ice crystal number concentrations (ICNC) up to four orders of magnitude higher than INP number concentrations Luke et al. (2021). Field campaigns such as the Mixed-Phase Arctic Cloud Experiment (M-PACE)





(Zhao et al., 2021), the Ny-Ålesund AeroSol Cloud Experiment (NASCENT) (Pasquier et al., 2022) and the Aerosol-Cloud Coupling And Climate Interactions in the Arctic (ACCACIA) (Sotiropoulou et al., 2021) offer robust observational evidence of the occurrence of SIP in Arctic clouds, not only due to large differences between ICNC and INP number concentrations, but also due to the presence of fragments of frozen drops, needles and sheaths as well as broken dendrite branches in images obtained from in-cloud sampling systems (Rangno and Hobbs, 2001; Young et al., 2016; Pasquier et al., 2022).

The three most studied SIP processes relevant for mixed-phase clouds are: 1) rime splintering also known as the Hallett-Mossop process, 2) droplet shattering during freezing and 3) ice-ice collisional breakup. Fragile ice crystals like dendrites may break up mechanically when colliding with another large ice particle (Takahashi et al., 1995; Phillips et al., 2017). The process is effective at cooler temperatures around -15 °C. When the surface of a large drizzle droplet freezes, e.g. by contact with an ice particle, the resulting pressure increase within the droplet may cause the ice surface to break, which releases small

ice fragments, or the droplet may eject an ice particle (Phillips et al., 2018; Keinert et al., 2020). Recent observations (e.g., Keinert et al., 2020) have shown that the process can take place at temperatures well above -10 °C but then the droplet needs to be large to produce significant number of secondary ice particles. The most studied SIP process is rime splintering (Hallett and Mossop, 1974) where fragile heavily rimed ice particles release ice splinters when colliding with large drizzle droplets, although the exact mechanism is not well known (Seidel et al., 2024). The process is most effective at temperatures close to

-5 °C.

In this study we performed ten meter-resolution large-eddy simulations (LESs) using the University of California Los Angeles Large Eddy Simulation model combined with the two-moment cloud microphysics scheme of Seifert and Beheng (2006) (UCLALES-SB) and with the Sectional Aerosol module for Large Scale Applications cloud microphysics scheme (UCLALES-SALSA) to investigate the interplay between primary and secondary ice production processes, which can determine the phase

and longevity of Arctic mixed-phase clouds. Our LES study is based on observations from the ACLOUD (Arctic CLoud Observations Using airborne measurements during polar Day) campaign carried out at north-west of Svalbard (Norway) in May-June 2017 (Ehrlich et al., 2019). Due to the observed conditions showing the absence of large drizzle drops and ice particles and cloud top temperatures close to -5 °C, we will focus on the rime splintering process. Our first aim is to examine what kind of adjustments (if any) are needed for the rime splintering parametrization to reproduce the observed ice concentration. Then we

will examine the impacts of meteorological and modelling uncertainties on secondary ice production. We will also examine if the results are sensitive on microphysics by comparing two-moment and sectional representations. Overall, our study aims to quantify the potential of Hallett-Mossop process in representing secondary ice production in such warm mixed-phase clouds.

## 2 Methods

### 2.1 The ACLOUD campaign

Current LES simulations are based on observations from the Arctic CLoud Observations Using airborne measurements during polar Day (ACLOUD) campaign (Ehrlich et al., 2019). The campaign included airborne observations from the Arctic boundary layer and clouds to understand their roles in Arctic amplification. Low-level clouds were frequently observed during a warm



and moist period from May 30 to June 12, 2017 (Wendisch et al., 2019). Here we focus on three flights with the Polar 6 aircraft conducted on the 2nd, 4th, and 5th of June when mixed-phase clouds were observed. These observations and the data analysis

are described in detail by Järvinen et al. (2023), so a summary focused on the model simulations is given here. During the three flights, there was a uniform non-precipitating cloud deck above pack ice in a region North of Svalbard (the current focus region is 8.5–12.0 °E and 81.1–81.4 °N). Our simulations will be focused on the 2nd of June flight, which is the one with the highest observed ice crystal concentrations and cloud top temperatures, but we will use the two other flights to assess the impact of variability of meteorological conditions.

As reported by Järvinen et al. (2023), cloud liquid (LWP) and ice (IWP) water paths were in the range of 48–82 $\mathrm{g\,m^{-2}}$ and 4.1–9.5 $\mathrm{g\,m^{-2}}$, respectively, for the three flights. Cloud base heights were between 100 and 200 m while the cloud top height was consistently about 440 m. The maximum cloud droplet number concentrations (CDNCs) from average vertical profiles (Järvinen et al., 2023, Fig. 6) varied between 66 and 152 $\mathrm{cm^{-3}}$. These correlate with the above-cloud aerosol concentrations ranging from 125 to 173 $\mathrm{cm^{-3}}$. Measured droplet size distributions showed the absence of large drizzle droplets. Namely, the

largest liquid droplets were about 30 μm in diameter (Supplementary material in Järvinen et al. (2023)). The observed cloud top temperatures ranged from -6.7 to -4.6 °C. Figure A1a shows the observed temperatures along with the idealized initial profiles for the model simulations (see Sect. 2.3).

Non-spherical ice particles in the diameter range from 9 to 1550 μm were detected using three different instruments as explained by Järvinen et al. (2023). Particle shattering due to collisions with the probes had an impact on concentrations

at the lower part of the cloud, and if such shattering was observed then ice particles smaller than 200 μm were excluded. The maximum 10 s average ice crystal number concentrations at the upper part of the cloud varied between 10 and 18 $\mathrm{L^{-1}}$ (Järvinen et al., 2023) while the mean was close to 1 $\mathrm{L^{-1}}$, which is in line with other observations from that region (e.g., Mioche et al., 2017). The ice particle shape observations showed that most particles were single crystals including needles and columns. In addition, significant fraction (38.5 %) of the observed ice crystals were rimed.

Ice-nucleating particles (INPs) are needed to initiate primary ice formation at the observed cloud temperatures (Kanji et al., 2017). There were no airborne INP measurements, but some ship-based measurements are available although only at -22.5 °C temperature (Wendisch et al., 2019). Even at such a low temperature, measured daily (June 2–5, 2017) INP concentrations are in the order of 0.1 $\mathrm{L^{-1}}$. These measurements rule out possible pollution episodes and indicate that low INP concentrations can be expected for the region of interest. The best literature estimates for INP concentration at the cloud top temperature of about

-5 °C is in the order of $1\times10^{-3}$ $\mathrm{L^{-1}}$ (e.g., Kanji et al., 2017; Murray et al., 2021; Li et al., 2022). This is significantly less than the observed cloud ice concentration of about 1 $\mathrm{L^{-1}}$ (Järvinen et al., 2023, Fig. 7). The three orders of magnitude difference between the INP and ice concentrations indicates that there is at least one active SIP process. The existence of large rimed ice crystals and the -5 °C cloud temperatures indicate that rime splintering (Hallett and Mossop, 1974) and droplet shattering during freezing (Phillips et al., 2018; Keinert et al., 2020) could be the dominating SIP mechanisms. However, droplet shattering

is limited to large drizzle droplets which were not observed, so rime splintering is the most likely SIP process.





## 2.2 UCLALES-SALSA

UCLALES-SALSA (Tonttila et al., 2017; Ahola et al., 2020) is the LES model used in this study. SALSA refers to the sectional aerosol-cloud microphysics, which was added to UCLALES as an additional module. UCLALES (Stevens et al., 1999, 2005; Stevens and Seifert, 2008) with the default "SB" two-moment bulk microphysics by Seifert and Beheng (2001) is a commonly used LES model especially for liquid clouds (e.g., Ackerman et al., 2009; Seifert and Heus, 2013; van der Dussen et al., 2013). In this study we also use the more recently updated two-moment bulk ice microphysics (Seifert and Beheng, 2006; Seifert, 2008; Seifert et al., 2012, 2014; Blahak, 2008; Noppel et al., 2010), which is also used in large scale models (Seifert et al., 2012; Hohenegger et al., 2023). For clarity, this model version is referred to as UCLALES-SB. Both model versions share the same LES framework including radiative transfer and surface interactions and only their microphysics differ.

Computationally light but simplified SB microphysics allows conducting hundreds of simulations, which is useful for tasks like sensitivity tests where the impacts of model parameters on predictions is quantified. SALSA microphysics allows explicit modelling of aerosol-cloud-ice processes but this comes with a significant computational cost. By comparing predictions from both SB and SALSA microphysics, we can see the impact of the level of microphysical details.

### 2.2.1 SB microphysics

Liquid clouds in UCLALES-SB are diagnostic, which means that cloud water mixing ratio is diagnosed by using the saturation adjustment approach and a fixed cloud droplet number concentration is specified as a model input. The two-moment rain microphysics by Seifert and Beheng (2001) describe both total mass and number concentrations while size distribution is assumed to follow a fixed gamma-distribution. Rain drop formation is based on a autoconversion parametrization and then the droplets can grow by condensation of water vapour and by collecting cloud droplets and smaller rain drops. The liquid-cloud scheme was extended for mixed-phase and ice clouds by Seifert and Beheng (2006); Seifert (2008); Seifert et al. (2012, 2014). The solid particle types include ice, snow, graupel, and hail, which have both mass and number as prognostic variables. The two-moment scheme accounts for various interactions between liquid and solid particles. The details are given in the original publications, so only a brief description is given here. The ice category represents small ice crystals formed by ice nucleation that are growing mainly by deposition of water vapour. In the absence of prognostic aerosols and thus INPs, primary ice nucleation is parametrized, so that the in-cloud ice crystal number concentration depends only on temperature. Collisions of ice with cloud droplets and larger rain drops leads to rimed ice, and depending on the resulting particle size, those particles are described by the snow, graupel and hail categories. Further riming and accretion lead to even larger particles and the resulting type is determined based on the size and type of colliding particles.

In this study we use hydrometeor parametrizations (fall velocity–mass–dimension parametrizations, parameters of the gamma-distribution, and mass limits) from Seifert et al. (2012) with the exception that fall velocity–mass–dimension parametrization for ice which is from Seifert et al. (2014). This change was made because the mass–dimension parametrization of ice from Seifert et al. (2012) produces exceptionally high dimensions compared with those from any other parametrization used in our simulations. Järvinen et al. (2023) used mass–dimension parametrizations from Brown and Francis (1995) to calculate the





ice water path (IWP) from the measured particle size. This parametrization, which happens to be the same as the Seifert and
Beheng (2006) snow parametrization, gives the same mass for $1.2\,\text{mm}$ particles as the parametrization by Seifert et al. (2014).
For particles smaller than $1.2\,\text{mm}$, the Brown and Francis (1995) parametrization gives higher mass than the parametriza-
tion by Seifert et al. (2014). Because simulated particles are typically smaller than $1.2\,\text{mm}$, the Brown and Francis (1995)
parametrization gives smaller dimension than that from Seifert et al. (2014). This will be examined in Sect. 3.4.

The only SIP process included is rime splintering (Eq. 1). Splinter production rates ($dN_i/dt$, $\text{s}^{-1}$) are parametrized as
product of constant $350\,\text{mg}^{-1}$ giving the number of splinters per milligram of rime, temperature-dependent efficiency term
$f(T)$, and water mass riming rate $dm_{\text{rime}}/dt$ ($\text{kg}\,\text{s}^{-1}$) (e.g. Hallett and Mossop, 1974; Cotton et al., 1986; Reisner et al.,
1998). The efficiency is linear between the minimum (zero at $265\,\text{K}$), optimal (one at $268\,\text{K}$) and the maximum (zero at $270\,\text{K}$)
temperatures. We assume that splinters are small and therefore assign them to the ice category.

$$dN_i/dt = 350\,\text{mg}^{-1}\,f(T)dm_{\text{rime}}/dt \tag{1}$$

This parametrization includes rime mass ($dm_{\text{rime}}/dt$) from collisions between any liquid droplet and solid ice particle.
Notably, any limits for droplet diameter such as $25\,\text{µm}$ minimum (e.g., Ferrier, 1994; Sullivan et al., 2018a) are excluded as
it would require calculating incomplete gamma functions. Also, the size limits are more important for parametrizations where
the number of splinters depends on the number of droplets collected (Field et al., 2017). Some studies have also limitations for
particle types that can produce splinters, for example, Kudzotsa et al. (2016), Sullivan et al. (2017, 2018b), and Sotiropoulou
et al. (2021) exclude ice particles while Sullivan et al. (2018a) include those. In this case, ice happens to be the dominant frozen
particle type, so excluding it would essentially prevent secondary ice production.

UCLALES-SB has mass concentration and the average maximum dimension thresholds for all collisions including riming,
which have no physical meaning but presumably were used to reduce computational costs. For example, the default minimum
total ice water mixing ratio and dimension are $10^{-5}\,\text{kg}\,\text{m}^{-3}$ and $150\,\text{µm}$, respectively, for ice-cloud collisions (the corre-
sponding limits for cloud are $10^{-6}\,\text{kg}\,\text{m}^{-3}$ and $10\,\text{µm}$). A dimension of $150\,\text{µm}$ is not a real limitation for the currently used
mass–dimension parametrization of ice. However, $10^{-5}\,\text{kg}\,\text{m}^{-3}$ is a high value considering the low primary ice concentra-
tion and the slow depositional growth rates. Thus, for all simulations here, we set the solid particle concentration limits to
$10^{-9}\,\text{kg}\,\text{m}^{-3}$, which is the same as the threshold concentration for rain. Atlas et al. (2020) made the same conclusion on con-
centration limits when simulating cumulus clouds over the Southern Ocean. Likewise, Schäfer et al. (2024) reduced thresholds
so that rime splintering could happen in their simulations with Morrison et al. (2009) microphysics based on the Ny-Ålesund
Aerosol Cloud Experiment (NASCENT). Similar adjustments have been made by Huang et al. (2008), Young et al. (2019), and
Sotiropoulou et al. (2021).

### 2.2.2 SALSA microphysics

Cloud microphysics in UCLALES-SALSA is treated using a sectional (bin) approach where aerosol, cloud, rain, and ice
are described using several size sections (bins) for which microphysical processes are calculated. The liquid and ice cloud
microphysics are originally described by Tonttila et al. (2017) and Ahola et al. (2020), respectively. The bins keep track of





chemical composition (mass of solutes and water) and the number of aerosol particles and hydrometeors. Aerosol and cloud droplet size bins are based on the dry particle size, which includes solutes but not water and assumes a spherical particle shape. Rain droplet size bins are based on the wet size that accounts for the droplet volume including solutes and water. For ice we also use liquid water-equivalent wet size bins which are independent of the assumed ice particle shape. The wet size is basically the same as the size of a liquid droplet resulting in from ice being melted. Water is allowed to partition between vapour, liquid and ice phases based on equilibrium conditions at the droplet (so-called $\kappa$-Köhler; Petters and Kreidenweis (2007)) and ice particle surfaces. Water vapour flux is diffusion-limited and related to ambient saturation ratio, thus this non-equilibrium approach allows the prediction of supersaturation and cloud activation without any additional parametrizations. Here cloud activation means that when the wet size of an aerosol bin exceeds the critical droplet size, it is moved (partially or completely) to a corresponding cloud bin. Rain drop formation can be based on either autoconversion-like bulk parametrization from SB microphysics (e.g., Seifert and Beheng, 2001) or counting the cloud-cloud collision where the resulting droplet size exceeds a threshold often set to 20 microns (Tonttila et al., 2021). Independent of the origin, rain drops will grow mainly by colliding with smaller cloud and rain droplets and eventually precipitate if conditions are suitable. Because liquid precipitation was not observed and simulated rain water paths were negligible, we will use the simple autoconversion-like bulk parametrization. For this study, we implemented the same rime splintering parametrization as used in the SB microphysics.

SALSA microphysics has certain concentration limits for all processes including riming, but the limits represent numerical accuracy of the model. Additional size limits are available for calculating the riming rate for the rime splintering process. The limit was set to $10\,\mu$m for cloud droplets, rain drops and ice particles. This means that the smallest cloud droplet bins can be excluded while all rain drops and ice particles are typically larger than $10\,\mu$m. When the size and particle type limits for riming and rime splintering are essentially removed, both SALSA and SB microphysics have similar chances to produce secondary ice particles.

## 2.3 LES setup

The LES domain covers a horizontal area of $10\,$km $\times\,10\,$km and extends vertically up to $1\,$km. Horizontal resolution is $100\,$m and vertical resolution is $10\,$m below $600\,$m. Above $600\,$m, vertical resolution increases by 3 % for each vertical level. Simulations have a maximum time step of $1\,$s and the total simulation time is 24 hours where the first hour is with liquid clouds only (spin-up). The spin-up is used to allow the development of turbulence in a liquid cloud before particle sedimentation and rain and ice microphysics are fully included. For short-wave radiation, the solar zenith angle is fixed to $60°$ to match with the observations made during 1–2 hours around the local noon in early June. In addition, following Järvinen et al. (2023), sea surface albedo is set to 0.5, which represents partial ice cover over pack ice. Long-wave emissions are based on the surface temperature, which is set to be the same as that of the initial atmospheric profile at the lowest model level. For pack ice we set the surface roughness to $0.04\,$m (see, e.g., Weiss et al., 2011). Latent and sensible heat fluxes are set to 15 and $0\,$W m$^{-2}$, respectively, based on initial tests where the fluxes were simulated. Large scale subsidence is described with a constant divergence of $5\times10^{-6}\,$s$^{-1}$.





Statistics are calculated every 2 minutes, and this is the output frequency of domain mean statistics. Horizontally and time-averaged profiles, and vertically integrated instantaneous column outputs are saved every 10 minutes.

For SB microphysics, we set the base case CDNC to $80\times10^6\,\mathrm{kg}^{-1} \approx 100\times10^6\,\mathrm{m}^{-3}$, which is in the range of the observed maximum values from $66\times10^6$ to $152\times10^6\,\mathrm{m}^{-3}$ (Järvinen et al., 2023). For SALSA we assume ammonium sulfate aerosol (hygroscopicity parameter $\kappa$=0.6) so that the shape of the initial unimodal log-normal size distribution ($D_g$=106 nm and

$\sigma$=1.81) is based on observations (see Fig. A2b) and the total aerosol concentration is set to $150\times10^6\,\mathrm{kg}^{-1}$. With this initial aerosol, simulated CDNC will be similar with the fixed value of $80\times10^6\,\mathrm{kg}^{-1}$ used by the SB microphysics. Aerosol is described with 12 logarithmically distributed dry size bins from 10 nm to 3 µm. Cloud bins are based on aerosol bins, but they do not include the first three bins (nucleation mode) as these particles are too small to activate. Rain bins range from 50 to 2000 µm and ice bins from 10 to 2000 µm (water-equivalent wet diameter). For SALSA ice, we use the same fall velocity–

mass–dimension parametrizations from Seifert et al. (2014) as with UCLALES-SB.

Because we are focusing on SIP, we use a simple primary ice formation approach where the in-cloud INP concentration is given as an input value. In practice, this means that cloud droplets freeze until the total ice concentration reaches the given INP concentration. As explained in Sect. 2.1, the expected INP concentration can be as low as $10^{-3}\,\mathrm{L}^{-1}$ or about $1\,\mathrm{kg}^{-1}$ (based on literature) but should not be larger than $0.1\,\mathrm{L}^{-1}$ or about $100\,\mathrm{kg}^{-1}$ (observed at -22.5 °C). So, with these LES simulations,

we will test different INP concentrations including 1, 10 and $100\,\mathrm{kg}^{-1}$. These INP concentrations are from one to three orders of magnitude lower than the observed ice concentration of about $1\,\mathrm{L}^{-1}$ or about $1000\,\mathrm{kg}^{-1}$ (Järvinen et al., 2023). We will also conduct a simulation where INP concentrations is set to $1000\,\mathrm{kg}^{-1}$ and SIP is switched off. This represents a modelling approach where high ice concentrations can be reached even without SIP by using unrealistically high INP concentrations.

The initial temperature and humidity profiles were reconstructed based on the observed cloud extent, LWP, and cloud top

temperature while also noting that these can change during the simulations. For example, ice formation and precipitation decrease LWP. The default profiles have specified surface temperature, relative humidity (RH) and pressure set to 1027 hPa, which allow the calculation of liquid water potential temperature ($\theta_L$) and total water mixing ratio ($r_t$) at the surface. These are assumed to be constant throughout a well-mixed boundary layer. A linear water vapour mixing ratio and potential temperature jumps (-0.9 g kg$^{-1}$ and 8 K, respectively) are assumed for the inversion layer (from 380 to 470 m), and above that the change

is based on fixed gradients of -0.001 g kg$^{-1}$ m$^{-1}$ for $r_t$ and 0.01 K m$^{-1}$ for $\theta_L$. Figure A1a shows the observations along with the absolute temperatures calculated from these profiles (based on the saturation adjustment method). Additionally, Fig. A1b shows the observed and average wind components, which are used in all LES simulations.

The two additional initial profiles (cool and moist) were generated to test the impact of observational variability of the meteorological parameters. The cool profiles were generated by decreasing $\theta_L$ by 2 K and by decreasing $r_t$ so that LWP is not

changing. The moist profiles were generated by increasing $r_t$ by 0.08 g kg$^{-1}$. In this case, the latent heating within the cloud layer increases the absolute temperature compared with that of the default profile. Figure 1 shows the initial temperature and moisture profiles as well as absolute temperature and RH based on the saturation adjustment method.



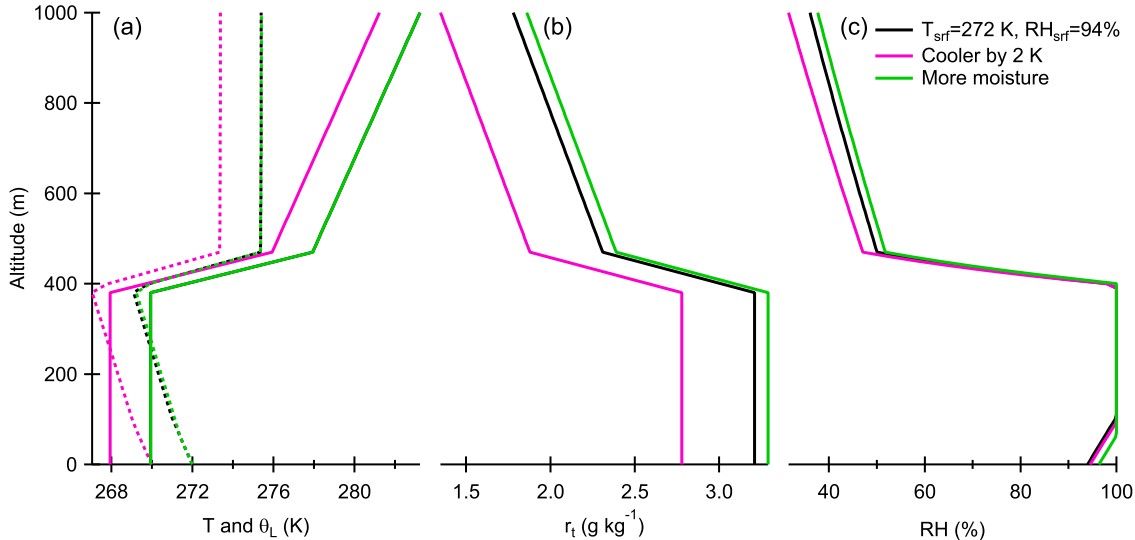

**Figure 1.** Initial (a) liquid water potential temperature ($\theta_L$, K, solid lines) and (b) total water mixing ratio ($r_t$, g kg$^{-1}$) profiles for the LES simulations including the base case and the cool and moist cases. Panel (a) shows also the absolute temperature ($T$, K, dashed lines) and panel (c) shows RH (%), which were calculated from $\theta_L$ and $r_t$ by using the saturation adjustment method.

# 3 Results

## 3.1 Base case

The first test simulations with UCLALES-SB showed that SIP was inefficient even after releasing the constraints related to droplet size, mass concentration or type of hydrometeors involved in the formation of rimed ice. Young et al. (2019) showed that in addition to removing size and concentration limits, secondary ice production had to be artificially increased to have an impact on ice concentration. So, to match with the observed ice crystal number concentration of about $1000\,\text{kg}^{-1}$, secondary ice production was increased by multiplying rime splintering rate (Eq. 1) by a constant factor while the temperature-dependent

efficiency term was retained. Figure 2 shows UCLALES-SB simulations with different INP concentrations (1, 10, 100, and $1000\,\text{kg}^{-1}$) when rime splintering (RS) rates are multiplied by a constant factor (0, 1, 5, and 10). Panel (a) shows the domain mean ice crystal number concentration (ICNC) for grid cells containing ice and panel (b) shows the horizontally averaged liquid water path (LWP). It should be noted that the SB microphysics includes ice, snow, graupel and hail categories, but only ice is shown here and in the following figures. This is because ice concentration is typically about two orders of magnitude

higher than the concentration of any other solid particle type.

The target (observed) ICNC of $1000\,\text{kg}^{-1}$ is reached when INP concentration is set to that value and rime splintering is switched off by setting the multiplier to zero (the one dashed line). Simulations with more realistic INP concentrations of 1, 10 and $100\,\text{kg}^{-1}$ and RS switched on (the unit multipliers) produce ice concentrations that are practically the same as the INP concentration, which means that secondary ice production is insignificant. Increasing the secondary ice production by a



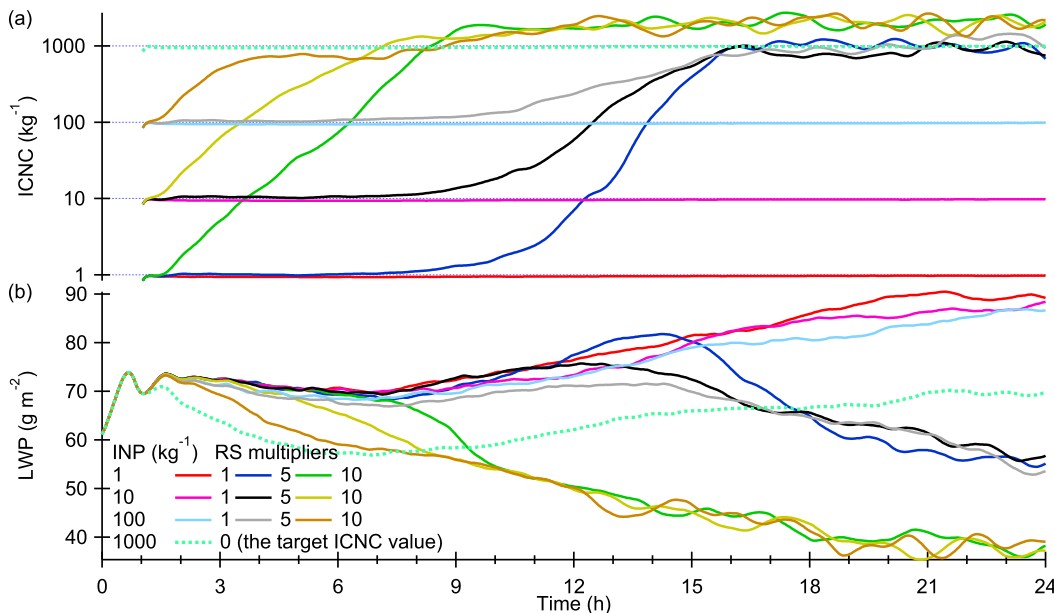

**Figure 2.** Simulated (a) ice crystal number concentration (ICNC) and (b) liquid water path (LWP) for the cases with different INP concentrations (1, 10, 100, and $1000 \, \text{kg}^{-1}$) and multipliers for the rime splintering (RS) secondary ice production rate (0, 1, 5, and 10) from UCLALES-SB.

factor 5 does help, but it requires about 15 hours until the target ice concentration is reached. This is a long time compared to, for example, diurnal temperature variations which could trigger or prevent secondary ice production. Increasing the rate by a factor 10 means that SIP starts almost immediately after the spin-up and the target ice concentration is reached within 9 hours. Interestingly, the factor of ten increase is enough for INP concentrations ranging from 1 to $100 \, \text{kg}^{-1}$ to reach the same steady-state ice concentration of about $2000 \, \text{kg}^{-1}$. Basically, this means that a strong enough SIP becomes self-sustaining, so

it no longer needs or depends on the primary ice formation. The same behaviour is seen in the simulations with a factor of 5 increase, but there is a significant time delay and the steady-state ice concentration is lower (about $1000 \, \text{kg}^{-1}$). Clearly, the more efficient SIP is able to maintain a higher steady-state ICNC.

When ice concentration is in the order of $1000 \, \text{kg}^{-1}$ (with or without SIP), the cloud starts to precipitate ice, and the continuous ice production and removal causes the decrease in LWP. This decrease in liquid cloud water reduces SIP but has no

direct impact on the INP concentration. Thus, cloud properties are different depending on how ice formation is modelled: with high INP concentration without SIP or low INP concentration and SIP producing most ice particles. Naturally, SIP accounts for the feedback between ice production and precipitation as well as any other removal mechanisms.

From now on, the factor of 10 increase in ice production is considered as the base case setting for SB microphysics. Interestingly, to match with the observed ice concentrations, Young et al. (2019) needed to apply the same factor of ten adjustment

to their rime splintering parametrization. Sotiropoulou et al. (2020) noted that when only the rime splintering process is ac-




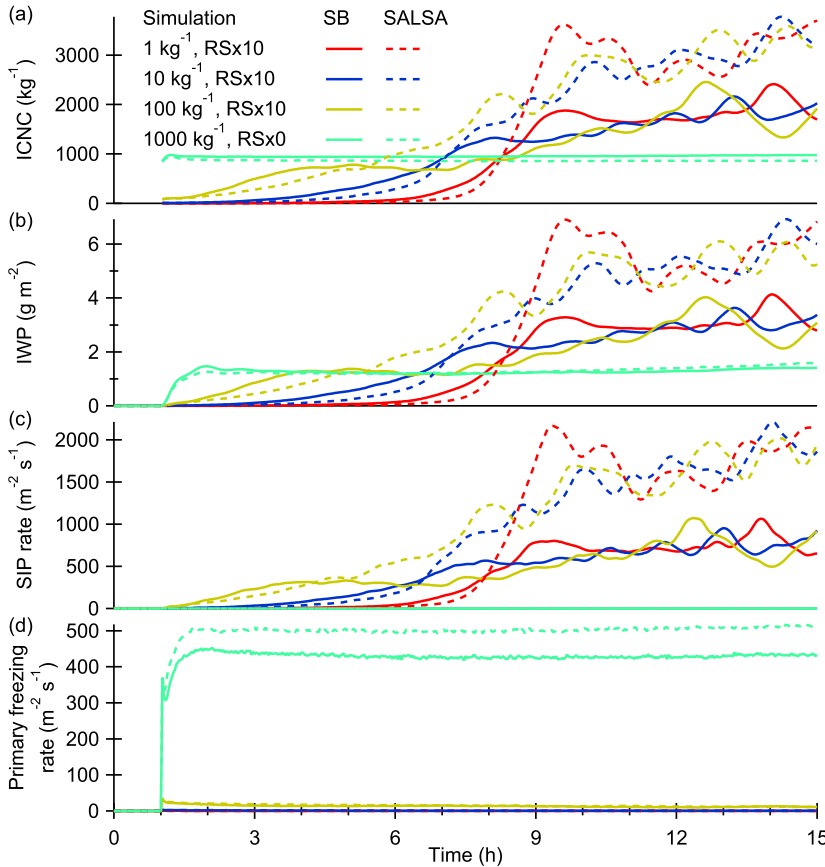

**Figure 3.** Simulated (a) ice crystal number concentration (ICNC), (b) ice water path (IWP), (c) secondary ice production rate, and (d) primary cloud droplet freezing rate for the cases with different INP concentrations (1, 10, 100, and $1000 \, \text{kg}^{-1}$) and multipliers for the rime splintering (RS) secondary ice production rate (0 or 10).

counted for, ice production had to be increased by about a factor of 10–20 to obtain a good agreement with the observed ice concentrations.

## 3.2 Comparison of cloud microphysical schemes

Figure 3 shows additional domain mean statistics from the four UCLALES-SB simulations (solid lines) described above and

from the corresponding UCLALES-SALSA simulations (dashed lines). The reference simulations have INP concentration set to $1000 \, \text{kg}^{-1}$ and SIP is switched off (RS×0). The three SIP simulations with INP concentrations set to 1, 10 and $100 \, \text{kg}^{-1}$ have the rime splintering (RS) ice production rate multiplied by ten (RS×10). Time in this figure is limited to 15 hours, because this is enough for both SB (see Fig. 2) and SALSA simulations to reach a steady-state.





The first thing that Fig. 3 shows is that initially SB has higher SIP rates and ice concentrations, but SALSA has higher
steady-state SIP rates so that the final ICNC is about $3000\,\mathrm{kg}^{-1}$ while this for SB this is $2000\,\mathrm{kg}^{-1}$. Overall, however, SB
and SALSA predictions are qualitatively similar, which is not always seen when comparing microphysical models that differ
so much in complexity. In this case, the complexity influences the time needed to run a simulation: about 1 h 20 min with
the two-moment SB and 29 hours with the sectional SALSA (parallel run with 100 CPU cores), i.e., there is a factor of 20
difference in computational costs. Clearly, the efficiency of SB makes it useful for conducting large numbers of test simulations
while SALSA can provide additional details about the process.

Another thing that Fig. 3 confirms is that SIP rate (panel (c)) exceeds the primary ice production rate (panel (d)) within
two to eight hours depending on the INP concentration. When ice concentration becomes large enough (about $1000\,\mathrm{kg}^{-1}$),
contribution from primary freezing becomes negligible. Thus, SIP maintains a feedback loop where INPs are not needed any
more. This was confirmed by a test simulation, where switching off the primary freezing after 6 h had negligible impact.

The third thing that Fig. 3 reveals is that the SIP rate correlates linearly with ice crystal number concentration and ice water
path (IWP). Indeed, calculating spatial and temporal correlations between SIP rate and various model outputs reveals that the
highest absolute Pearson's correlation coefficients are seen for ice number concentration, water vapour deposition rate, and
IWP. Table 1 shows Pearson's correlation coefficients for these variables (and LWP as a reference) calculated for both SB and
SALSA simulations where the INP concentration is set to $100\,\mathrm{kg}^{-1}$ and SIP rate is multiplied by a factor of 10. Temporal
correlation is calculated for the domain mean time series outputs and spatial correlation is calculated for snapshots of column-
averaged or integrated 2D model outputs taken from the 7[th] hour. These three independent, i.e., not directly related to the SIP
rate like riming rate, variables clearly stand out. ICNC, water vapour deposition rate, and IWP represent the 1[st], 2[nd] and 3[rd]
moment of the ice size distribution, respectively. The most obvious explanation is that SIP requires cloud droplet–ice collisions.
Because cloud droplets and LWP are more evenly distributed, ice crystals are more important for the spatial correlation. The
negative temporal correlation between SIP rate and LWP is related to the fact that ice is produced at the expense of liquid
water, which is also apparent from Fig. 2. Luke et al. (2021) found a positive spatial correlation between observed vertical air
velocity and SIP rates, but this is not that clear in our simulations, because the correlation coefficients (0.36 for SB and 0.25
for SALSA) are smaller than those for LWP. In fact, it looks like higher vertical velocities mean higher LWPs, which support
ice production.

As an example, Fig. 4a shows the temporal correlation between domain mean SIP rate and IWP for all SB and SALSA
simulations where SIP is enabled. Figure 4b shows a snapshot of vertically integrated SIP rate contours over IWP colourmap
(SB simulation, INP=$100\,\mathrm{kg}^{-1}$, RS$\times$10, time=7 h). Clearly, the SIP rate contours match with the regions with high IWP.

## 3.3 Vertical distributions

Comparing horizontally averaged profiles of the cloud parameters (Fig. 5) shows fairly small differences especially when each
profile is selected so that the ICNC reaches $1000\,\mathrm{kg}^{-1}$ for the first time (here we take the concentration at the altitude of
355 m). Panel (c) shows that the fixed CDNC for the SB microphysics matches well with the prognostic CDNC from the
SALSA simulations. This is the case because total aerosol number was adjusted for this purpose. The most obvious difference



**Table 1.** Spatial and temporal correlation between SIP rate and the three independent variables with the highest Pearson's correlation coefficients (Deposition rate, IWP, and ICNC) and additionally also LWP. Spatial correlations are calculated for the column-averaged model outputs taken from hour 7 while temporal correlation is calculated for the domain mean output time series. All simulations have INP concentration set to $100\,\mathrm{kg^{-1}}$ and SIP rate multiplied by a factor of 10.

|  | Spatial | | Temporal | |
| --- | --- | --- | --- | --- |
| Variable | SB | SALSA | SB | SALSA |
| LWP | 0.60 | 0.55 | -0.77 | -0.93 |
| ICNC | 0.82 | 0.92 | 0.91 | 0.97 |
| IWP | 0.90 | 0.94 | 0.90 | 0.95 |
| Deposition rate | 0.93 | 0.97 | 0.92 | 0.96 |

is that SIP produces ICNC profiles that have a maximum within the cloud and lower values below while the profiles without SIP are essentially uniform and even increasing with decreasing altitude. The ACLOUD observations cannot show if the profiles should be uniform or not. This is because the observations are subjected to the typical detection limits (cannot see the smallest particles) as well as particle shattering effects (Järvinen et al., 2023), which impacts depend on altitude.

Figure 6 shows additional statistics about vertical distributions. Panel (a) shows the minimum absolute temperatures. These and especially their minimum values (cloud top temperatures) are similar for all simulations. Panel (b) show that the parametrized primary ice nucleation takes place mostly at the cloud top, but for the no-SIP simulation ($1000\,\mathrm{kg^{-1}}$, RS$\times$0) the rates are significant for the whole cloud layer. SIP rates are distributed more evenly based on the cloud temperature and liquid water mixing ratio (Fig. 5d). The rime splintering process takes place between 265 and 270 K and the maximum efficiency is at 268 K, which is lower than the minimum cloud top temperatures. In addition, the maximum rate is seen at about 400 m altitude (high liquid water mixing ratios) where the temperatures are about 269 K. This indicates that a cooler temperature profile where the minimum temperature is slightly below 268 K would increase SIP rates. This will be examined in the next section. Due to the different distributions of the primary and secondary ice production, ice crystals in the SIP runs are larger (panel (d)) at the altitudes below the SIP region. There is also a difference between SB and SALSA microphysics so that the mean diameter is larger in SB simulations.

### 3.4 Sensitivity tests

Here we conduct sensitivity tests based on both observational and model variables that are most influential for SIP. We use the same approach as above, i.e., determine a multiplier for the SIP rate so that the simulated ice concentration match with the observed ice concentrations of about $1000\,\mathrm{kg^{-1}}$. These simulations are made with the computationally light SB microphysics, because as shown above, SALSA produces qualitatively similar results. We will focus on the case with the highest INP concentration of $100\,\mathrm{kg^{-1}}$ thus the base case simulation has SIP rate multiplied by ten (RS$\times$10). Here we limit simulation time to 10 hours.



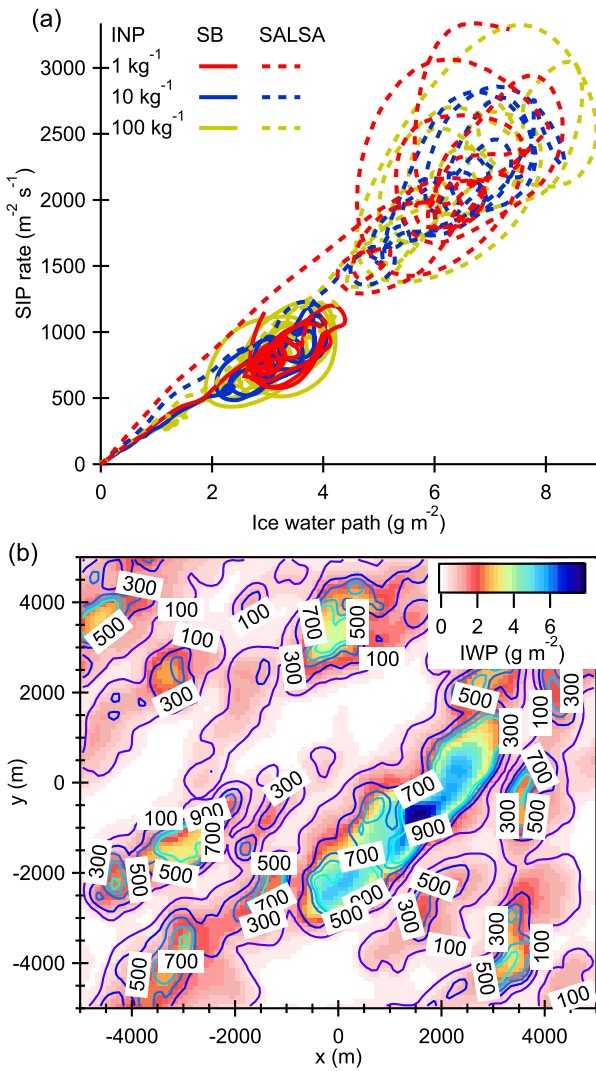

**Figure 4.** Simulated (a) domain mean SIP rate as a function of IWP for the six simulations and (b) instantaneous ($t$=7 h) SIP rate contours (the change in column ICNC due to SIP, $m^{-2}\,s^{-1}$) and IWP colourmap for the SB simulations with INP=$100\,kg^{-1}$ and SIP rate multiplied by a factor of 10.

The observational variables that will be examined include CDNC and cloud temperature and water content (LWP). Cloud temperature has a direct impact on the rime splintering process while CDNC and LWP have an impact on cloud dynamics. Figure 7 shows how these observational uncertainties influence secondary ice production. When simulations are initialized with the humid total water mixing ratio profile (see Fig. 1), LWP is about 30 % higher, but this has a relatively small impact on ice concentration (the same RS multiplier as in the base case can be used). The cool profile ($\theta_L$ reduced by 2 K), on the

other hand, has a clear impact on SIP. Just a factor of two multiplier is enough to start significant ice production. Initially, the



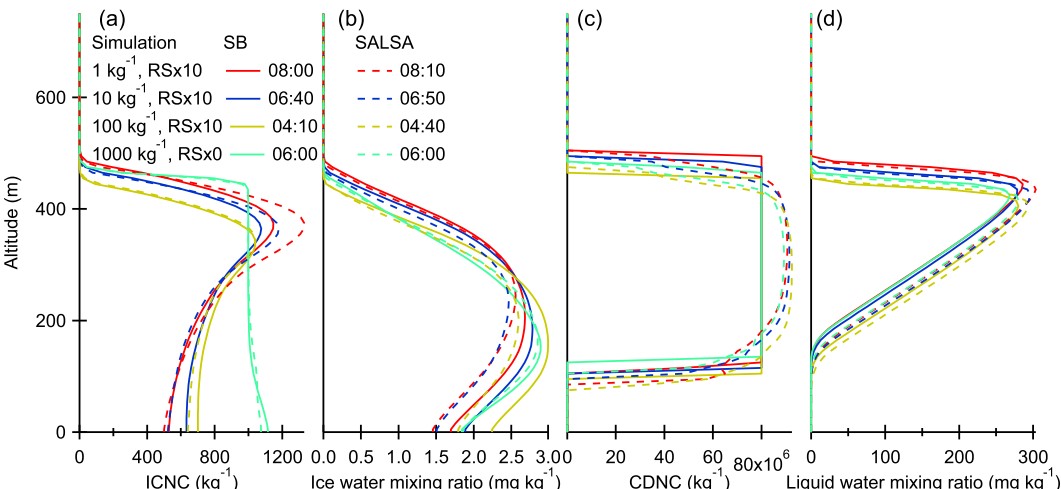

**Figure 5.** Profiles of (a) ICNC, (b) ice water mixing ratio, (c) CDNC, and (d) liquid water mixing ratio from the different simulations. The time (hh:mm) for each simulation is selected so that ICNC first exceeds $1000\,\mathrm{kg}^{-1}$ at the altitude of 355 m.

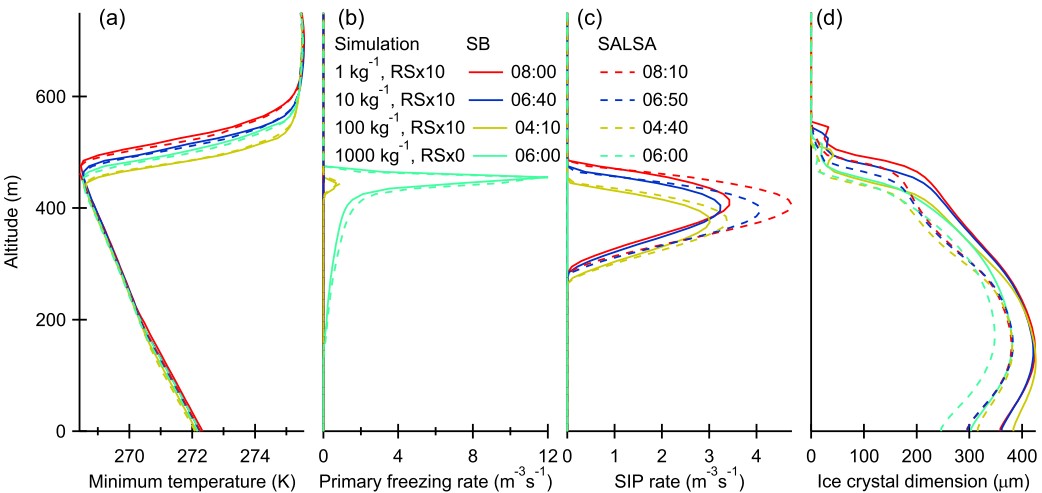

**Figure 6.** Profiles of (a) minimum absolute temperature, (b) primary freezing rate, (c) secondary ice production rate, and (d) ice crystal dimension from the different simulations. The time (hh:mm) for each simulation is selected so that ICNC first exceeds $1000\,\mathrm{kg}^{-1}$ at the altitude of 355 m.

ice concentration increases rapidly but soon the increased precipitation removal starts to limit ice production. Reducing CDNC from $80\times10^6\,\mathrm{kg}^{-1}$ by 50 % to $40\times10^6\,\mathrm{kg}^{-1}$ means that cloud droplets are larger, which means higher fall velocities, so also the riming rate increases. As a result, reducing the multiplier by 40 % from 10 to 6 is enough to reach and overpass the target ICNC of $1000\,\mathrm{kg}^{-1}$ around the 6$^\text{th}$ hour.



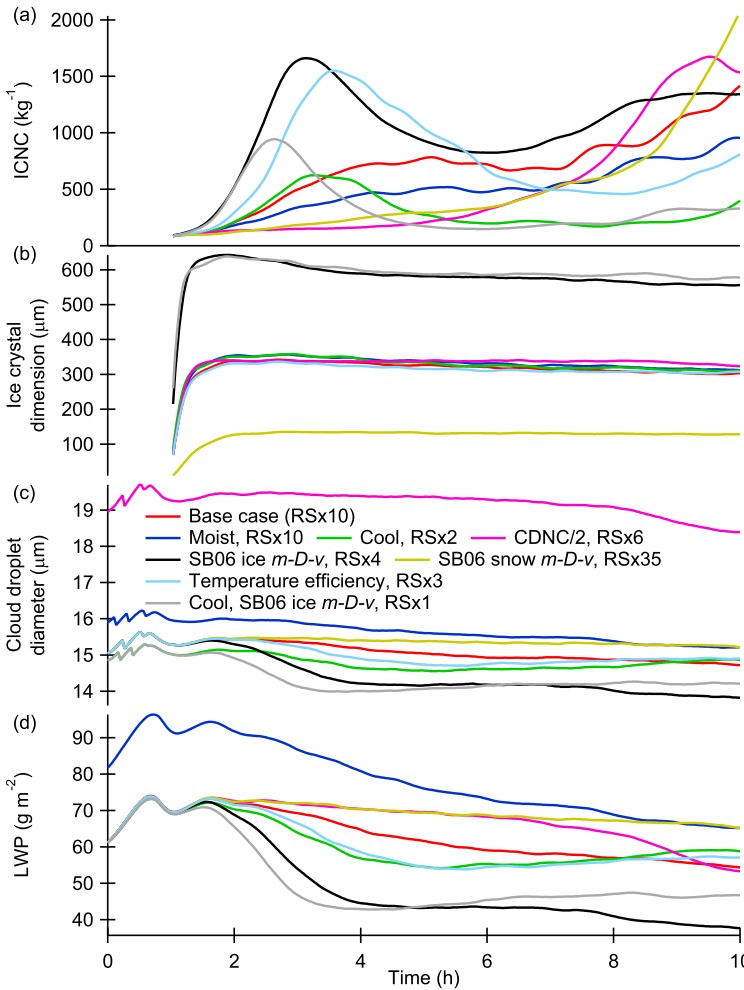

**Figure 7.** Sensitivity tests based on the observed variability of moisture (Moist), temperature (Cool) and cloud droplet number concentration (CDNC/2), mass-dimension-velocity ($m$–$D$–$v$) parametrizations from Seifert and Beheng (2006) (SB06 ice and snow $m$–$D$–$v$), and temperature efficiency ($f(T)$ in Eq. 1). The last test is with cool profiles and SB06 ice $m$–$D$–$v$ parametrization. In each simulation the SIP rate is multiplied by a factor so that ice concentration increases to about $1000\,\text{kg}^{-1}$. INP concentration is $100\,\text{kg}^{-1}$ in all simulations.

Modelling uncertainties are also significant and not only related to the rime splintering parametrization (the number of fragments per accumulated mass of rime, temperature limits, and possible size limits). From the many adjustable model parameters, mass-dimension-velocity ($m$–$D$–$v$) parametrizations seem to have the largest impact on SIP. In Fig. 7, we show simulations where the current ice parametrization is replaced by ice and snow parametrizations from Seifert and Beheng (2006), SB06. The SB06 ice parametrization represents an extreme parametrization regarding ice crystal size, which is increased by almost 100 %

(from about 300 to 600 µm). Also, the fall velocity changes, but this has negligible impact on SIP. With this parametrization, SIP becomes more efficient so that a factor of four increase for SIP rate is enough. The SB06 snow parametrization includes the





same $m$–$v$ parametrization as is used for the current ice, so we only change the $m$–$D$ parametrization which is the same as used by Järvinen et al. (2023) (from Brown and Francis (1995)) for calculating ice mass from the observed ice crystal shapes. This parametrization drastically reduces particle radius from about 300 to $100\,\mu m$. This reduced SIP so that it must be multiplied

by a factor of 3.5 in addition to the original 10. The importance of the $m$–$D$ parametrization can be understood by the fact that collision kernel, which is used for calculating riming rate, is related to the square of the dimension while fall velocity has only linear dependency.

The currently used triangular temperature efficiency curve (linear from zero at $265\,K \approx$ -8 °C to one at $268\,K \approx$ -5 °C and back to zero at $270\,K \approx$ -3 °C) is less efficient compared with some other alternatives. For example, Sotiropoulou et al. (2020)

used piecewise constant efficiency curve so that it is one for -6 °C $< T <$ -4 °C and 0.5 for the other temperatures between -8 °C $< T <$ -2 °C (Ferrier, 1994). Sullivan et al. (2018b) have unit efficiency between -8 °C $< T <$ -3 °C and 0.01 elsewhere (Takahashi et al., 1995). Ziegler et al. (1986) has parabolic efficiency for temperature range from -2 to -8 °C. As a temperature efficiency test, we modified the efficiency so that it is one between -8 and -3 °C and zero elsewhere. This increases SIP so that only a factor of three increase is needed for the rime splintering.

Overall, this sensitivity study suggests that with the cooler temperature profiles, slightly lower CDNC, the ice $m$–$D$–$v$ parametrization from SB06, and the more efficient temperature dependency, the LES can reproduce the observed ice concentration of about $1000\,kg^{-1}$ without modifying the rime splintering parametrization, and indeed this is the case. This is shown by the last sensitivity test in Fig. 7. Here we have cooler temperature profile and use SB06 ice $m$–$D$–$v$ parametrization, but some other combinations of the adjustments would have the same effect.

## 4  Conclusions

Here we used observations by Järvinen et al. (2023) to initialize LES simulations that aimed at reproducing the high ice concentrations observed in a relatively warm mixed-phase cloud deck where secondary ice production (SIP) was expected to dominate over the primary freezing initiated by INPs. Cloud temperatures were about -5 °C, so we focused on rime splintering also known as the Hallet-Mossop process. With the default microphysical setup the model was not able to produce secondary

ice, even after giving up from the commonly applied size and particle type limitations, so we increased the rime splintering SIP rate by a constant factor. A factor of ten increase was required for the base case so that SIP was able to first increase the ice crystal number concentration (ICNC) from the primary ice concentration as low as $1\,kg^{-1}$ to the observed value of about $1000\,kg^{-1}$, and then maintain that over several hours. Basically this means that a strong enough SIP can become self-sustaining and thus be independent on the primary freezing. Interestingly, the factor of ten increase worked well for the two

cloud microphysics models used in this study: the detailed sectional SALSA and the fast two-moment SB (Seifert and Beheng, 2006). The factor of ten happens to be the same enhancement as used in some previous studies (Young et al., 2019; Sotiropoulou et al., 2020; Schäfer et al., 2024).

An alternative for artificially adjusting SIP rate is adjusting temperature efficiency or other model parametrizations (mass–dimension–fall velocity) or setup (temperature) to increase SIP. The triangular temperature efficiency curve (linear from zero at





265 K ≈ -8 °C to one at 268 K ≈ -5 °C and back to zero at 270 K ≈ -3 °C) used in the current rime splintering parametrization is less efficient compared with some other alternatives. For example, Sotiropoulou et al. (2020) used piecewise constant efficiency curve so that it is one for -6 °C < $T$ < -4 °C and 0.5 for the other temperatures between -8 °C < $T$ < -2 °C (Ferrier, 1994). Sullivan et al. (2018b) had unit efficiency between -8 °C < $T$ <-3 °C and 0.01 elsewhere (Takahashi et al., 1995). Using any one of these would increase SIP rates. Moisture content has a much smaller effect while cloud droplet number concentration (CDNC) can have a significant effect especially when CDNC is low so that cloud droplets become larger. Suitable combination of those can easily initiate self-sustaining secondary ice production. On the other hand, defining size or particle type limits may completely prevent SIP for certain cloud types, especially shallow clouds that have relatively small ice particles and narrow range of in-cloud temperatures. Ideally, such conditions should be replaced by smooth probability terms that reduce ice production in the case of unfavourable conditions. Overall, our results support the previous findings about the high sensitivity of SIP on various model setups and environmental conditions, which is a challenge for modelling.

For the shallow clouds in this study, the other potential SIP mechanism is droplet shattering (DS) based on the temperature range, although large droplets (>30 µm) were not observed (Järvinen et al., 2023). The DS and RS parametrizations share similar features, both being dependent on ice–droplet collisions. For example, Lawson et al. (2015) matched model simulations accounting for droplet shattering to observed ice concentrations by adjusting the number of secondary ice particles produced by per kilogram of accumulated rime mass, which is analogous to the rime splintering parametrization but with different temperature dependency. The optimal value was reported as $10^{-9} \, \mathrm{kg}^{-1}$, but presumably this should be $10^{9} \, \mathrm{kg}^{-1}$, which is about three times larger than the corresponding constant for RS parametrization. However, Lawson et al. (2015) derive this value by assuming that DS is the only SIP process when at least ice-ice collisional breakup would be active at the simulated temperatures between -8 and -20 °C. Due to the similarity, it is not surprising that either DS (Lawson et al., 2015) or RS alone can produce high ice concentrations. This is probably one reason why RS parametrizations are often successfully used as the only SIP parametrization, including this study. It can't be argued that the current RS parametrizations have weaknesses that need to be addressed and that the other SIP processes (at least droplet shattering and ice-ice collisional breakup) should be accounted for when conditions are more suitable for them. We will focus on these processes in another study (Calderón et al., 2025) where cloud temperatures are lower and thus more favourable for them

Regardless of the exact mechanism, it is essential to account for SIP rather than fix INP or ice crystal number directly. The simulations showed that vertical ice distributions and ice crystal sizes are different depending on how they were simulated. Moreover, using SIP allows the negative feedback between precipitation and ice production, which allows the development of stable mixed-phase clouds. Increase in the ice concentration will deplete liquid water, which in turn reduces SIP rates and stabilizes the cloud phase partitioning. With high fixed ice concentrations, the clouds are more likely to dissipate or glaciate, which is an issue seen in many large-scale models. Although the simulated vertical ice profiles were different with and without SIP, it was not possible to see if the observations match better with either one of those. However, this could be possible in future studies.

A recent study (Seidel et al., 2024) has questioned the existence of the rime splintering process. Our study cannot confirm that the process is real, but at least the simulated ice concentrations match well with the observed concentrations. The currently





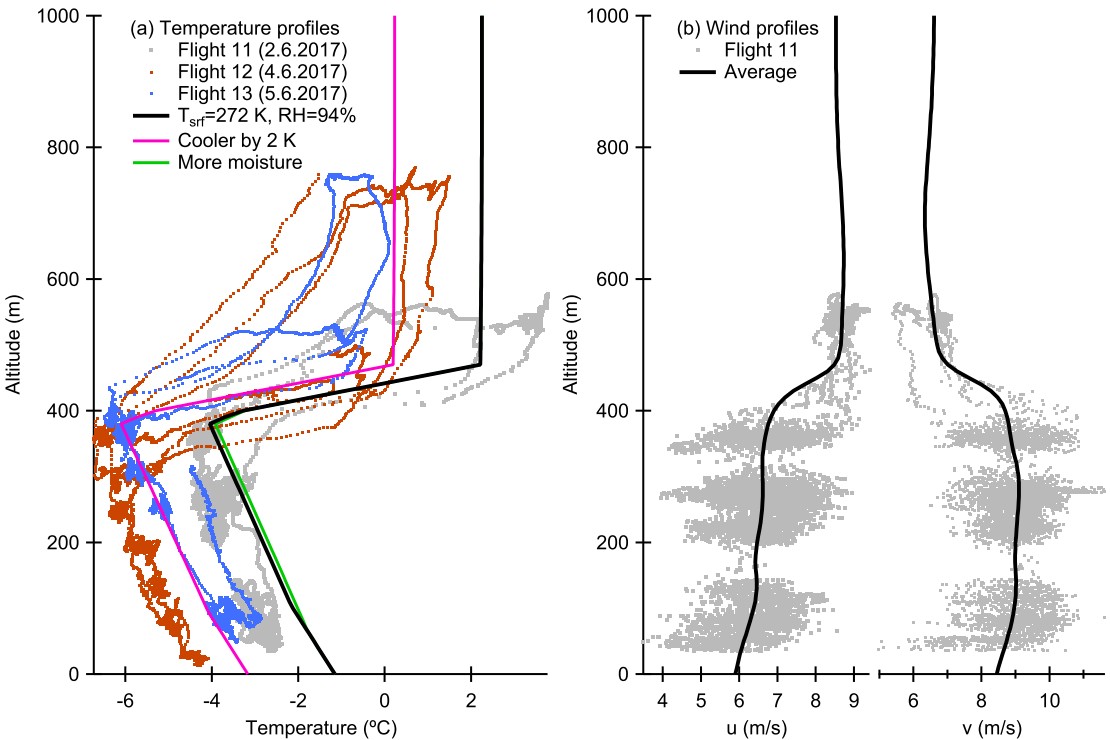

**Figure A1.** Observed (a) temperature profiles from three research flights during the warm period, and (b) wind profiles for flight 11 (Hartmann et al., 2019). The solid black lines indicate the default LES initialization based on observations from flight 11 (June 2, 2017). The cool and moist temperature profiles are used for additional sensitivity tests described in the main text.

(and commonly) used parametrization is simple enough for models that have simplified microphysics (e.g., large-scale models) so it is useful at least for now.

## Appendix A:  Simulation settings

Figure A1a shows the observed temperatures from three research flights during the warm period and Fig. A1b shows wind speed components from flight 11 (Hartmann et al., 2019). The solid black lines indicate the default LES initialization based on

observations from flight 11 (June 2, 2017). To account for the radiative cooling, the minimum temperature seen at the cloud top is set to be slightly warmer than the observed minimum temperature of -4.56 °C. The initial rapid cooling decreases simulated minimum temperatures to -4.5 °C and then the cooling continues at a slower rate of -0.03 °C/h. The initial wind profiles were calculated from the observations as a weighted mean. The weight for altitude $x$ (based on the LES grid) for wind velocity observation at altitude $z_i$ is $w_i(x) = \exp(-((x - z_i)/(50\,\mathrm{m}))^2)$.



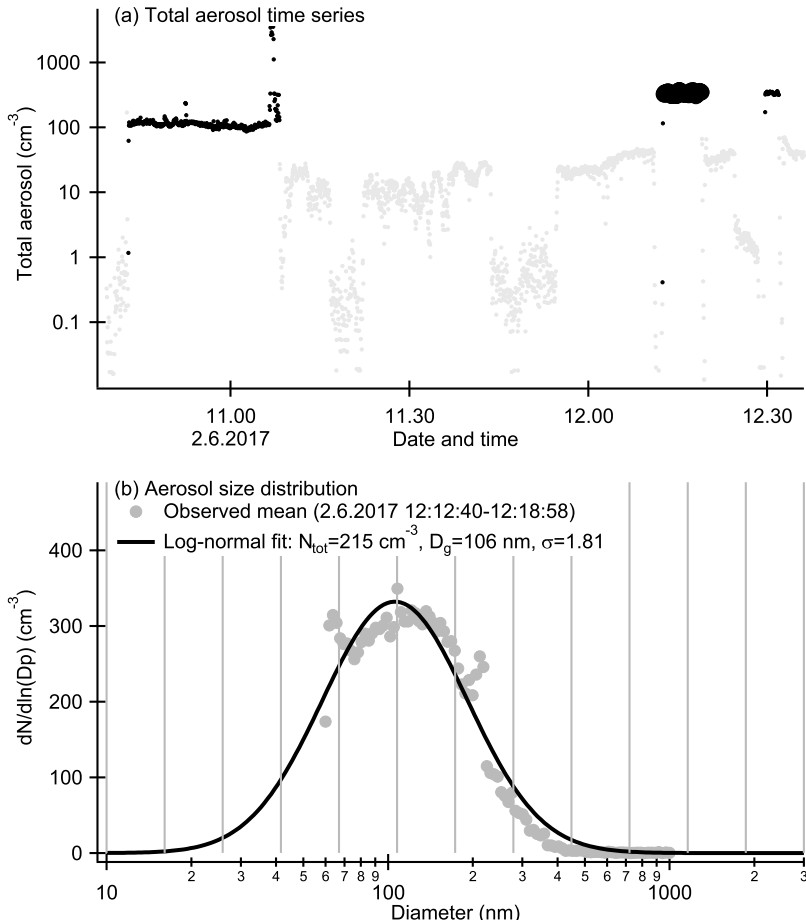

**Figure A2.** Observed (a) total aerosol number concentration time series and (b) the average ambient aerosol size distribution. The data is from June 2, 2017, flight (Mertes et al., 2019). The black and grey colours in panel (a) indicate time periods when measuring ambient aerosol and cloud particle residuals, respectively. The ambient aerosol size distribution in panel (b) is averaged from time period 12:12:40 – 12:18:58, which is marked with the larger dots in panel (a). Panel (b) also shows a log-normal fit to the data covering the SALSA aerosol bins (bin limits indicated by the grey vertical lines).

Figure A2a shows the observed total aerosol number concentration and Fig. A2b shows the average ambient aerosol size distribution (Mertes et al., 2019). The average aerosol size distribution includes observations from time period 12:12:40 – 12:18:58 when sampling ambient aerosol (marked with the larger dots in panel (a)). The log-normal fit in panel (b) was used to initialize aerosol size distribution for SALSA except that the total aerosol number concentration was set to $150 \times 10^6 \, \text{kg}^{-1}$ ($195 \, \text{cm}^{-3}$ when air density is $1.3 \, \text{kg m}^{-3}$) so that the simulated CDNC matched with the observed value of about $80 \times 10^6 \, \text{kg}^{-1}$.

Table A1 shows the model parameters and settings for all LES simulations. These are also described in the main text.



**Table A1.** Model parameters and other simulation settings.

| Parameter | Common defaults |
|---|---|
| Horizontal domain | Nx=Ny=100, dx=dy=100 m |
| Vertical domain | Nz=85, dz=10 m below 600 m and stretched by 1.03 above that |
| Time | Maximum step=1 s, total=86400 s, spin-up=3600 s |
| Outputs | Statistics every 120 s, averages every 600 s |
| Mean winds | u=9.0 m s$^{-1}$, v=6.0 m s$^{-1}$ |
| $\Theta_{00}$ | 270 K |
| Four-stream radiation | SZA=60°, background=mid-latitude winter atmosphere (kmlw.lay), $\alpha$=0.5, SST=272 K |
| Large-scale divergence | $5\times10^{-6}$ s$^{-1}$ |
| Surface | $z_0$ = 0.04 m, SHF=0 W m$^{-2}$, LHF=15 W m$^{-2}$ |
| Primary ice | Cloud droplets freeze when $S_i$>0 and ICNC<INP concentration |
| INP concentrations | 1, 10, 100, and 1000 kg$^{-1}$ |
| RS temperature efficiency | $f(T)$ is linear between $T_{min}$=265 K, $T_{opt}$ = 268 K, and $T_{max}$ = 270 K |
| Ice $v = a_v * m^{b_v}$ | $a_v$=27.7, $b_v$=0.216 |
| Ice $D = a_D * m^{b_D}$ | $a_D$=0.835, $b_D$=0.390 |
|  | **SB defaults** |
| Fixed CDNC | $80\times10^6$ kg$^{-1}$ |
| Riming limits | Droplets: $r_c$>$10^{-6}$ kg m$^{-3}$, $D_c$>10 μm |
|  | Ice: $r_i$>$10^{-9}$ kg m$^{-3}$, $D_i$>0 μm |
|  | **SALSA defaults** |
| Initial aerosol | Log-normal size distribution: $N_{tot}$=$150\times10^6$ kg$^{-1}$, $D_g$=0.106 μm, $\sigma$=1.81 |
|  | Composed of sulfate: M=132.14 g mol$^{-1}$, $\rho$ = 1770 kg m$^{-3}$, $\nu$ = 2.49 |
| Riming limits | Droplets: $N_c$>$10^{-3}$ m$^{-3}$, $D_c$>10 μm |
|  | Ice: $N_i$>$10^{-6}$ m$^{-3}$, $D_i$>10 μm |
| Aerosol bins | 12 logarithmically spaced bins between 10 and 3000 nm |
| Rain bins | 7 logarithmically spaced bins between 50 and 2000 μm |
| Ice bins | 10 logarithmically spaced bins between 10 and 2000 μm |
|  | **Sensitivity tests** |
| Moist | Moist initial profiles |
| Cool | Cool initial profiles and SST=269.985 K |
| CDNC/2 | CDNC=$40\times10^6$ kg$^{-1}$ |
| SB06 ice $m$–$D$–$v$ | $a_v$=317, $b_v$=0.363, $a_D$=0.217, $b_D$=0.302 |
| SB06 snow $m$–$D$–$v$ | $a_v$=27.7, $b_v$=0.216, $a_D$=8.156, $b_D$=0.526 |
| RS temperature efficiency | $f(T)$=1 between 265 and 270 K |





*Code and data availability.* Brief description of the simulations, source code of UCLALES-SALSA, and the simulation data used in this publication are available from https://a3s.fi/12001823-acloud-pub/index.html [for review] (Raatikainen, 2025).

*Author contributions.* TR designed and conducted UCLALES-SALSA simulations. TR, SC, MP, and SR have contributed to developing the UCLALES-SALSA model. EJ provided the observational data used in this study. TR prepared the manuscript with contributions from all

co-authors.

*Competing interests.* The authors have no competing interests to declare.

*Acknowledgements.* The authors wish to acknowledge CSC – IT Center for Science, Finland, for computational resources. This research has been supported by the Research Council of Finland (decision numbers 322532 and 359342) and by the European Union's Horizon Europe CleanCloud (grant agreement no. 101137639) and CERTAINTY (no. 101137680) projects.



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
