# Peer review of "Can rime splintering explain the ice production in Arctic mixed-phase clouds?"

_EGUsphere, 2025_

## Referee Comment (RC1)

This modeling study examines whether the Hallett Mossop process can explain the high ICNCs observed within an Arctic stratocumulus deck during ACLOUD campaign, using LES simulations. Their results show that the current widely-used parameterization for the Hallett-Mossop mechanism cannot reproduce observations, unless its efficiency is enhanced by factor of ten. Alternatively, other microphysical formulations (e.g. terminal velocity calculation or HM temperature efficiency) should be modified in certain ways that can positively impact ice production from this process, to achieve a realistic ICNC representation.

The large discrepancies between INP-ICNC observations in Arctic clouds have been documented for a long time, with SIP being hypothesized as the potential cause. However, atmospheric models still struggle to reproduce the microphysical structure of Arctic clouds, even when SIP parameterizations are accounted for. This is largely due to the fact that SIP descriptions remain highly uncertain, which is why sensitivity studies like this one are very useful for the cloud modeling community.

Overall, I find the design of the study and the sensitivity tests satisfying, however I believe that results are not properly discussed. For example, while observations of micro- and macro-physical properties are available, they are not used for evaluation of e.g. different schemes or different HM formulations. Moreover, there is hardly any discussion on the differences in using different microphysical schemes (these sensitivity tests are neither discussed in the abstract nor in the conclusion section). I also think there some inconsistencies between the background information provided in this study and the relevant papers that are cited as reference.

A more detailed list of comments is given below. I don't think it would take a long time to the authors to address these - for this reason I consider the suggested revisions as minor.

**Main comments:**

Line 10: 'SIP depends strongly on model parametrizations', this is a very generalized statement (and not anything new). Should become more specific on what kind of parameterizations you refer to

Line 14: 'cooler marine regions', cooler than what

Line 20: 'simplified cloud microphysics.' Could you provide reference? I do not necessarily agree. I think two-moment fully prognostic schemes are rather complex

Line 32: 'The process is effective at cooler temperatures around -15 ° C'. Takahashi et al. (1995) with a rather unrealistic set-up showed maximum efficiency round -15 °C. However, numerous more recent studies have shown that this process can be active over a much wider temperature range

Line 34: Not true. Figure 7 in Keinert et al (2020) shows that the process can be effective above - 10°C. It is **more** effective below this temperature

Line 187-189: What was the criteria for adjusting surface fluxes and subsidence? These choices can have a large impact on the simulated LWP/TWP.

Line 87-89: I don't understand why the existence of large rimed ice crystals are only indicative for the Hallett-Mossop and drop-shattering process. The efficiency of collisional break-up highly depends on ice particle size and riming (Phillips et al. 2017)

Line 243-245: RSx10 overestimates ICNCs and underestimates LWP. This is not addressed in the discussion. Overall it would be helpful if all ICNC and INP units are the same throughout the whole text to facilitate comparison. E.g. observations are given in L-1, while all simulated values in kg-1.

Figure 2: It should also illustrate a timeseries of IWP to assess which simulation is more realistic compared to the measured values (4.1–9.5 g m–2). A shaded area in the individual panels that indicates the observed ICNC, LWP and IWP ranges would also be very useful for evaluation

Line 245-246: I don't understand why the RSx5 simulation is not considered more realistic since it better reproduces the observed ICNCs. It also gives more realistic LWP values within the observed range (48–82 g m–2). Could the time-delay of ICNC enhancement in this simulation be due to the short spin-up time? (not enough to allow cloud dynamics to develop in time, which are critical for many microphysical processes?)

Line 250-251: This statement is not clear; what is considered high and low INP value? 'Moderate' values also produce the desired ICNCs when combined with SIP

Line 253: Not clear. What do you mean by the term 'removal mechanism'? Do you refer to cloud condensate removal, ice mass or liquid mass removal? The different LWP values for example suggest different impact on liquid mass removal mechanisms.

Line 255-257: I am not sure to which sensitivity simulation in this study do you refer to. In their RS experiment they showed that the simulated ICNCs where 10-20 times lower the observed not necessarily that the ice production rate should be multiplied by 10 (which is the case with SI rate in Young et al). In their DM10\_SIP experiment indeed they multiplied primary ice production (DeMott et al 2010) by a factor of ten, but this simulation accounted for many SIP mechanisms (not just RS). The relevant statement should be more accurate to avoid misinterpretation. Moreover, they showed that the combination of RS with another SIP mechanism was essential to reproduce the observed ICNCs. Maybe along these lines or during the last section it should be discussed that the RS multiplier could actually account for the missing contribution of another SIP mechanism

Figure 3: It would be very useful to see how timeseries of LWP also differs between the two microphysical schemes. Again the observed range of values should be marked in the Figure to facilitate evaluation

Section 3.2: In my opinion the similarities between the two simulation set-ups is overemphasized, while there is hardly any discussion about the obviously statistically significant differences. ICNC is 35% larger (which is mentioned), however IWP is also twice as larger in SALSA. LWP might also reveal notable differences.

Line 283: I guess SIP rates are added to the ice crystal number tendency equation in the code. Why is it stated here that ICNCs are not directly related to SIP rate?

Line 316-317: I would also argue that SB despite being simplified, it agrees more with observations (ICNC, IWP, etc)

Figure 7: Timeseries of IWP would likely be useful too

381-382: I still do not understand why collisional break up is not considered a potential mechanism. Have you looked for fragmented ice particles in the proble images?

Lines 409-410: I don't understand why a better fit to the temperature measurements was not used for the LES intitialization. Clearly the authors preferred to adapt a warmer and more unstable profile.

Conclusions: results related to the different microphysics schemes are not discussed

**Typos:**

Line 320: water content (LWC)

Line 359: Hallett

---

## Author Comment (AC1)

**Reply to referee comments**

We would like to thank the two referees for their constructive comments, which helped us to improve the manuscript. The specific comments (highlighted in blue font) are shown below with our replies (normal text).

One issue that both referees raise is that other SIP mechanism (droplet shattering and ice-ice collisional breakup) used in our previous study (Calderón et al.: Secondary ice formation in cumulus congestus clouds: insights from observations and aerosol-aware large-eddy simulations, *Atmos. Chem. Phys.*, 25, 14479-14500, https://doi.org/10.5194/acp-25-14479-2025, 2025.) were excluded. In this study, we employed another model version that did not include the other SIP processes (but includes the Seifert and Beheng (SB) microphysics). Also, the temperature range and the absence of large droplets pointed towards rime splintering. Nevertheless, we have now implemented the missing ice-ice collisional breakup (IIBR) and droplet shattering (DS) processes to our SALSA version and repeated the simulations (please, see the revised manuscript). As expected, (1) rime splintering is the dominating SIP process and (2) including the other SIP processes does not change model predictions significantly, so our conclusions are valid.

Figure 1 shows results from the revised (default setup and RS multiplied by one and ten) and previous (RS multiplied by ten; no other SIP processes) model versions. Clearly RS is about three orders of magnitude stronger than the other processes. In addition, rime splintering production rates as well as ice crystal number concentrations (ICNCs) from the original and new RSx10 simulations are practically indistinguishable, which means that the other two SIP processes are insignificant. We have also verified that RS, IIBR and DS SIP rates are similar for the other two INP concentrations (1 and 10 $kg^{-1}$).

The manuscript is updated so that previously missing SIP processes are now described briefly and their impact on ice concentrations is quantified. All SALSA results (figures and tables) have been updated, because they now account for the three SIP processes, but they have no impact on our conclusions. Results based on the Seifert and Beheng (SB) microphysics have not changed.

[Figure]

Figure 1: Ice crystal number concentrations (ICNCs) and secondary ice production (SIP) rates from UCLALES-SALSA simulations with INP concentration of $100\,\mathrm{kg}^{-1}$ and different multiplication factors for the rime splintering (RS) process. The last simulation is from the old manuscript where RS is the only SIP process. The thick gray line shows the target minimum ICNC of $1000\,\mathrm{kg}^{-1}$.

**Referee #1**

REVIEW FOR RAATIKAINEN ET AL.

This modeling study examines whether the Hallett Mossop process can explain the high ICNCs observed within an Arctic stratocumulus deck during ACLOUD campaign, using LES simulations. Their results show that the current widely-used parameterization for the Hallett-Mossop mechanism cannot reproduce observations, unless its efficiency is enhanced by factor of ten. Alternatively, other microphysical formulations (e.g. terminal velocity calculation or HM temperature efficiency) should be modified in certain ways that can positively impact ice production from this process, to achieve a realistic ICNC representation.

The large discrepancies between INP-ICNC observations in Arctic clouds have been documented for a long time, with SIP being hypothesized as the potential cause. However, atmospheric models still struggle to reproduce the microphysical structure of Arctic clouds, even when SIP parameterizations are accounted for. This is largely due to the fact that SIP descriptions remain highly uncertain, which is why sensitivity studies like this one are very useful for the cloud modeling community.

Overall, I find the design of the study and the sensitivity tests satisfying, however I believe that results are not properly discussed. For example, while observations of micro- and macro-physical properties are available, they are not used for evaluation of e.g. different schemes or different HM formulations. Moreover, there is hardly any discussion on the differences in using different microphysical schemes (these sensitivity tests are neither discussed in the abstract nor in the conclusion section). I also think there some inconsistencies between the background information provided in this study and the relevant papers that are cited as reference.

A more detailed list of comments is given below. I don't think it would take a long time to the authors to address these - for this reason I consider the suggested revisions as minor.

**Main comments:**

Line 10: ' SIP depends strongly on model parametrizations', this is a very generalized statement (and not anything new). Should become more specific on what kind of parameterizations you refer to

Changed to 'SIP depends strongly on model parametrizations (e.g., fall velocity–mass–dimension parametrizations and those describing the dependency of SIP on temperature and particle size and habit)'.

Line 14: 'cooler marine regions', cooler than what

Changed cooler to high-latitude.

Line 20: 'simplified cloud microphysics.' Could you provide reference? I do not necessarily agree. I think two-moment fully prognostic schemes are rather complex

We removed 'simplified cloud microphysics' because low resolution alone is enough.

Line 32: 'The process is effective at cooler temperatures around -15°C'. Takahashi et al. (1995) with a rather unrealistic set-up showed maximum efficiency round -15°C. However, numerous more recent studies have shown that this process can be active over a much wider temperature range

We will clarify that the maximum efficiency is close to that temperature. We will also add simulations (please, see the beginning of this reply and the revised manuscript) where the three SIP processes are included to show that rime splintering is the dominant process in our case.

Line 34: Not true. Figure 7 in Keinert et al (2020) shows that the process can be effective above - 10°C. It is more effective below this temperature

Keinert et al. (2020) and the above -10 °C temperatures are already mentioned just below that line. As mentioned above, we tested the effectiveness of droplet shattering SIP.

Line 187-189: What was the criteria for adjusting surface fluxes and subsidence? These choices can have a large impact on the simulated LWP/TWP.

As mentioned in the text, we did simulations where the fluxes were calculated based on surface temperature (close to freezing point of sea water) and RH=100 % at the surface. The fixed surface flux values are based on those simulations. Divergence 5e-6 s$^{-1}$ may sound quite high, but for the low clouds in this study it means modest vertical velocity of about 2.5 mm/s at the cloud top (500 m altitude). Based on several test simulations, this value was found to be large enough to limit the increase in LWP in simulations with low ice concentration (less precipitation) and, on the other hand, low enough to maintain sufficiently high LWP in simulations with high ice concentration (more precipitation). Overall, the fluxes are small as expected and together with the subsidence they lead to stable clouds and especially ice concentrations.

Line 87-89: I don't understand why the existence of large rimed ice crystals are only indicative for the Hallett-Mossop and drop-shattering process. The efficiency of collisional break-up highly depends on ice particle size and riming (Phillips et al. 2017)

We removed this text because our new simulations show the effectiveness of the three SIP processes.

Line 243-245: RSx10 overestimates ICNCs and underestimates LWP. This is not addressed in the discussion. Overall it would be helpful if all ICNC and INP units are the same throughout the whole text to facilitate comparison. E.g. observations are given in L-1 , while all simulated values in kg-1.

ICNC is the only target value, but we will clarify that 1000 kg$^{-1}$ is the concentration that should be reached to achieve the closure with observations. The upper limit in observed ICNC is much higher (10 000 kg$^{-1}$), and it is never reached in our simulations. At the cloud top where ice particles smaller than 200 $\mu$m can be included in the total concentration, the profile average ICNC values are close to 10 L$^{-1}$ but the standard deviation is large ranging from detection limit (0.01 L$^{-1}$) up to 100 L$^{-1}$ (Fig. 7 in Järvinen et al., 2023). The same figure shows that when particles larger than 200 $\mu$m are excluded (due to observed shattering in the Particle Habit Imaging and Polar Scattering (PHIPS) and The Small Ice Detector Mark 3 (SID-3) probes), the mean is about 1 L$^{-1}$ with much smaller standard deviation. Based on these values, the best estimate for ICNC is between 1-10 L$^{-1}$ (range from 0.1 to 100 L$^{-1}$). In the original manuscript, we took the lower limit (1 L$^{-1} \approx 1000$ kg$^{-1}$) as the target value, which needs to be reached. We will clarify this in the revised manuscript.

LWP is an input value (related to the initial temperature and moisture profiles) for the free-running simulations where ice formation and precipitation can reduce LWP. In addition, the development of LWP depends on surface fluxes and subsidence, and in this case subsidence was adjusted so that LWP is not increasing too much in simulations with low ice concentration (see the comment above). The precipitation rate depends on ICNC and the onset time for SIP reaching significant levels. Thus, simulations with low ice concentrations agree with the initial LWP while simulations with realistic ice concentrations will have a lower LWP. Therefore, comparison with the observed LWP would give a wrong impression that LWP is too low when ice concentration is realistic.

We cannot fully avoid using different units, because the observational values are given in $L^{-1}$, but most model inputs (e.g., INP concentration) and outputs are mass-based. Instead of converting all model outputs (and inputs) to $L^{-1}$, we converted the observed ice concentrations to $kg^{-1}$. After all, this is a modelling study and the bias due to unit conversion (air density) is negligible for observational values that have large uncertainty ranges.

Figure 2: It should also illustrate a timeseries of IWP to assess which simulation is more realistic compared to the measured values (4.1–9.5 g m-2). A shaded area in the individual panels that indicates the observed ICNC, LWP and IWP ranges would also be very useful for evaluation

The observed IWP is based on ice water contents (IWC), which depends on an assumed mass-dimension (m–d) relationship. One m–d formulation gave the numbers mentioned above, but another formulation gave IWCs lower by an order of magnitude (Järvinen et al., 2023). Thus simulated IWP range from 1 to 7 g $m^{-2}$ is within the observational range including uncertainty. Because the observational uncertainty covers the range of predictions, it is not practical to add such wide shaded area to the figures. Comparison between LWP observations and model predictions is not meaningful for the reason mentioned above (LWP is a model input or initial value, not a target).

As mentioned above, we aim at reaching ICNC of about 1000 $kg^{-1}$, which corresponds to the lower limit of observations. The upper limit (10 000 $kg^{-1}$) is never reached, so basically all our simulations (RSx10) are within the observed range. We will add a line representing the target (minimum) value of 1000 $kg^{-1}$ to Figs 2 and 3.

Line 245-246: I don't understand why the RSx5 simulation is not considered more realistic since it better reproduces the observed ICNCs. It also gives more realistic LWP values within the observed range (48–82 g m-2). Could the time-delay of ICNC enhancement in this simulation be due to the short spin-up time? (not enough to allow cloud dynamics to develop in time, which are critical for many microphysical processes?)

As explained above, we aim at reaching ICNC of about 1000 $kg^{-1}$ and the LWP is model input rather than a target value. It is true that RSx5 is realistic (or any multiplier between 5 and 10), and we will clarify this in the revised manuscript. The RSx10 was selected to reduce the computational cost of the SALSA simulations.

One hour spin-up time is enough to allow the development of vertical mixing (the first increase in LWP). But we also did a simulation where longer (2 hours) spin up did not have significant effect on the ICNC trend. Basically, it just delayed the increase in ICNC by one hour.

Line 250-251: This statement is not clear; what is considered high and low INP value? 'Moderate' values also produce the desired ICNCs when combined with SIP

High INP concentration (without SIP) refers to the simulation with INP concentration set to the target [minimum] ICNC, and low means the other simulations where INP concentration is smaller than the ICNC. We will clarify this in the revised manuscript.

 Not clear. What do you mean by the term 'removal mechanism'? Do you refer to cloud condensate removal, ice mass or liquid mass removal? The different LWP values for example suggest different impact on liquid mass removal mechanisms.

SIP reduces when either cloud liquid or ice particles are removed. We reformulate this to 'cloud water and ice removal mechanisms such as precipitation'. Surely, ice-ice and ice-droplet collisions also change the amounts of cloud ice and liquid, but precipitation has more direct effect.

Line 255-257: I am not sure to which sensitivity simulation in this study do you refer to. In their RS experiment they showed that the simulated ICNCs where 10-20 times lower the observed – not necessarily that the ice production rate should be multiplied by 10 (which is the case with SI rate in Young et al). In their DM10_SIP experiment indeed they multiplied primary ice production (DeMott et al 2010) by a factor of ten, but this simulation accounted for many SIP mechanisms (not just RS). The relevant statement should be more accurate to avoid misinterpretation. Moreover, they showed that the combination of RS with another SIP mechanism was essential to reproduce the observed ICNCs. Maybe along these lines or during the last section it should be discussed that the RS multiplier could actually account for the missing contribution of another SIP mechanism

This line refers to this (Page 1312 in Sotiropoulou et al., 2020): "Our results indicate that the combination of both RS and BR is a possible explanation for the observed ICNCs; the newly generated fragments by RS further fuel the BR process, resulting in substantial ice enhancement through the latter, compared to when only one mechanism is active. Interestingly, when only RS is accounted for, the multiplication effect has to be increased by about a factor of 10–20 to obtain a good agreement with the observed ICNCs, i.e., the same factor as that used in Young et al. (2019)." It is not fully clear if they did a simulation where rime splintering was multiplied by those factors. It is true that they conclude that RS with another SIP mechanism was essential to reproduce the observed ICNCs. Our new simulations show that RS is the dominating mechanism, and RS (with or without the other SIP processes) is too weak when using the default model setup.

Figure 3: It would be very useful to see how timeseries of LWP also differs between the two microphysical schemes. Again the observed range of values should be marked in the Figure to facilitate evaluation

We will add LWP but the observed ranges are not included as explained above (LWP is an input).

Section 3.2: In my opinion the similarities between the two simulation set-ups is overemphasized, while there is hardly any discussion about the obviously statistically significant differences. ICNC is 35% larger (which is mentioned), however IWP is also twice as larger in SALSA. LWP might also reveal notable differences.

We surely expected that the two very different models will give different predictions, and the models do show statistically significant differences (e.g., 2000 vs 3000 $kg^{-1}$ for ICNC). This is much less than could be expected for this kind of case where SIP increases ice concentrations by orders of magnitude. The difference is also much smaller than the

range of observational values (at least an order of magnitude for ICNC). Since ICNC is essentially determining IWP, which is mostly reduced from LWP, also the IWP and LWP values differ. We will clarify that the predictions are similar considering the model differences and the case where SIP increases ice concentration by orders of magnitude.

Line 283: I guess SIP rates are added to the ice crystal number tendency equation in the code. Why is it stated here that ICNCs are not directly related to SIP rate?

The Hallett-Mossop parameterization does not depend on ICNC directly, but the parameterization is almost directly dependent on riming rate (Eq. 1 in the manuscript), which depends on both ICNC and cloud droplets. That's why there is no point in including riming rates in this discussion.

Line 316-317: I would also argue that SB despite being simplified, it agrees more with observations (ICNC, IWP, etc)

We cannot say which one is better, because observational uncertainties are much larger than the difference in model predictions.

Figure 7: Timeseries of IWP would likely be useful too

We will add IWP to the figure.

381-382: I still do not understand why collisional break up is not considered a potential mechanism. Have you looked for fragmented ice particles in the probe images?

As mentioned above, we ran simulations that include collisional breakup (and droplet shattering), and it was found much less effective than rime splintering.

The majority of the ice particles in the probe images were classified as needles or columns, and these habits were consistently imaged as intact crystals. In contrast, particles assigned to the "other polycrystal" category exhibit more complex morphologies, and it cannot be excluded that some particles in this class may represent fragments rather than intact crystals. However, the class covered typically <20% of all crystals.

Lines 409-410: I don't understand why a better fit to the temperature measurements was not used for the LES intitialization. Clearly the authors preferred to adapt a warmer and more unstable profile.

We will clarify the profiles are not directly fitted but are adiabatic (neutral/well-mixed) profiles below cloud top. The profile was selected so that the simulated cloud top temperature matches with the observed minimum value, which is important for SIP. The warmest case was selected simply because it is the most interesting for SIP (warm but high ice concentration) while the cooler and humid profiles were left to sensitivity tests. Overall, these setups cover the observed cloud top temperature and LWP ranges.

Conclusions: results related to the different microphysics schemes are not discussed

As explained above, we will clarify that the results are different but less than expected.

**Typos:**

Line 320: water content (LWC)

Changed to liquid water path (LWP)

Line 359 : Hallett

Fixed

**Referee #2**

This paper addresses the origin of the high ice particle concentrations observed in an Arctic mixed-phase cloud during the ACLOUD campaign, which is a topic that still raises several open questions. The main results show that the rime splintering (RS) mechanism alone cannot explain the observed ice crystal number concentrations (ICNC). The authors demonstrate that either increasing the RS efficiency by a factor of ten or modifying the model setup and parameterizations of microphysics can lead to more realistic ICNC values.

Overall, this is a good quality study. The combined use of both bin and bulk microphysics approaches is quite rare and strengthen the conclusions of the study. It is also very interesting to see that modifying the parameterizations of microphysics have a such large effect on SIP efficiency.

I see two main issues with the present study. First I don't understand why other SIP mechanisms are not considered, given that were studied in one previous study: Calderón et al. (2025) (https://doi.org/10.5194/egusphere-2025-2730). Even if the temperature range studied here is not optimal for a high efficiency of collisional breakup or fragmentation of freezing drops (more active around –15°C), these processes could still influence the ICNC. Secondly, while the sensitivity experiment are valuable, their design and presentation could be improved for greater clarity.

Despite these points, the study is of high quality and offers valuable insights into SIP. I recommend that the authors address the following points to further strengthen the manuscript before publication.

**Major comments:**

Although significant uncertainties remain regarding SIP, artificially increasing the rime splintering (RS) efficiency in the model seems a bit artificial, as the literature does not support such strong RS. This point should be more clearly emphasized in the study.

We fully agree that the increase is artificial as already mentioned in the text. This is simply the easiest way to reach targeted ice number concentration, which in this case is reasonable ice concentration ($1000 \text{ kg}^{-1}$) produced by SIP.

Collisional breakup the breakup process has been shown to be effective in Arctic mixed-phase clouds (e.g., Karalis et al., 2022: https://doi.org/10.1016/j.atmosres.2022.106302 and Sotiropoulou et al., 2021: https://doi.org/10.5194/acp-21-9741-2021). Therefore, it would also be valuable to test whether collisional breakup or DS could explain the observed ICNC without artificially boosting the RS rate. If I am correct, these SIP mechanisms can be activated in SALSA, as done in Calderón et al. (2025)?

Calderon et al. (2025) used different SALSA version, but we implemented their collisional breakup (IIBR) and droplet fragmentation (DS) parameterizations and repeated the simulations. As shown above (and explained in the revised manuscript), RS is three orders of magnitude more efficient compared with IIBR or DS. The studies by Karalis et al. (2022), Sotiropoulou et al. (2021) and Calderón et al. (2025) examine cooler and thicker clouds, where the conditions are more favourable for IIBR and DS.

As ice crystal concentration is typically expressed in $\text{L}^{-1}$, I recommend using this unit instead of $\text{kg}^{-1}$ for consistency with the INP units and the literature.

Instead of converting all model outputs (and inputs such as INP concentrations) to $\text{L}^{-1}$,

we converted the observed ice concentrations to $kg^{-1}$. After all, this is a modelling study and the bias due to unit conversion is negligible for observational values that have large uncertainty ranges.

Lines 69-70: Although no large drops were observed, it would be useful to check whether the model generates any. The presence of such drops could potentially trigger DS.

The model generates large (rain) droplets, but their concentration is negligible. For example, the highest rain drop number concentration in simulations shown in Fig. 3 is less than $2000\,kg^{-1}$. As mentioned above, we did simulations where DS and IIBR are included, but RS dominates.

Lines 73-79: The presence of millimeter sized ice particles suggest that collisional breakup might occur.

As mentioned above, we included collisional breakup, but it was found ineffective.

Lines 137-141: It is is good point to highlight and discuss that.

Now we mention excluding ice particles as an example in the conclusions.

Section 2.2.2: The HM process in SALSA and the number of bins used are not described, unlike for the SB scheme. Adding this information would improve clarity for readers.

We will clarify that the HM parameterization in SALSA is identical to that in SB (Eq. 1). SALSA bins were described in Sect. 2.3, but this information is now updated (the number of bins is given) and moved to Sect. 2.2.2.

Lines 238-239: This statement seems too strong. I would suggest saying that RS is insignificant, rather than SIP in general, since other SIP mechanisms were not considered.

Other SIP mechanisms are considered in the revised manuscript, but only with SALSA. These simulations show that rime splintering dominates. We have clarified the manuscript so that RS is now mentioned in relevant places.

Figure 2: In the factor 10 multiplier experiment, the LWP decreases to approximately 40 $g\ m^{-2}$, which is a bit lower than the observed values of 48–82 $g\ m^{-2}$ mentioned in Line 65. The factor 5 experiment appears to be more consistent with the observed range.

As explained above for Referee #1, LWP is used as model input (initial value) and not as target value. LWP decreases in these free-running (nudging not used) simulations depending mainly on ice concentration so that the decrease is larger with higher ice concentration. We also agree that the factor of five experiment produces reasonable ice concentrations.

Lines 255-257: The Sotiropoulou et al. (2020) study also mention that "*Only the combination of both rime splintering (RS) and collisional break-up (BR) can explain the observed ICNCs, since both of these mechanisms are weak when activated alone.*" This suggests that combining RS with other processes could give realistic ICNCs, rather than artificially increasing RS.

Other processes were tested, but RS is the most effective while the others have negligible impact. The other processes are important for cases with, e.g., thicker clouds or cooler temperatures.

Lines 267-269: It is good to highlight the computational cost of the two microphysics schemes.

This is now highlighted in the conclusions.

Section 3.3: for both Fig 5 and 6:

- The comparison between the 100 kg$^{-1}$ RS×10 line and the 1000 kg$^{-1}$ RS×0 line makes it difficult to determine whether the variations are due to turning off RS or changing the INP number. Adding an intermediate 100 kg$^{-1}$ RSx0 line would allow isolating the effect of RS deactivation without altering the INP number.

- Suggestion: showing only one time per profile could make the figure more readable, as these lines are similar and not central to the discussion.

Note that the intermediate (100 kg$^{-1}$ RS×0) line would differ very much from the other lines as ICNC in this case is about 100 kg$^{-1}$ while in the other cases it is 1000 kg$^{-1}$. Surely, an order of magnitude lower ice concentration would mean that at least the ice parameters differ.

One time per profile: does this mean removing either SB or SALSA profiles or removing something else like times that differ? Or using the same time for all? Different times are used because SIP simulations require different time (depends on INP concentration and microphysics) to increase ice concentration so that SIP takes control and the observed target ice concentration is reached. Comparison is fair when each profile is selected so that the target ice concentration is reached, i.e., different time for each profile.

Section 3.4: Examining the effect of the model parameters on SIP production is very interesting. Including a table in the appendix summarizing all the tests would help readers, or at least more clearly mentioning the experiment names in the text.

Note that Table A1 already summarizes the parameters used in sensitivity tests. We added a link to that table. We also added the experiment names to the text in bold font.

Fig 7: Presenting sensitivity results first with a fixed RS multiplier would help, as varying both RS and model parameters complicates comparisons. I suggest separating the experiments into (1) varying model parameters with RS fixed, and (2) varying the RS multiplier to clarify their effects.

That would mean showing many simulations where SIP is either ineffective or too effective. To keep the results for (2) readable, we would need one new figure for each test, so seven new figures in total, which is quite a lot. Current figure is a summary from such simulations showing only the relevant ones. Single figure is also better for comparing the relevant simulations.

Lines 340-342: Yes, this is an excellent point to highlight.

This is now highlighted in the conclusions.

Lines 343-349: This is very interesting. Why not test other temperature curves here as well?

The tested temperature curve represents a highly efficient one. Using a less efficient one would mean that the required RS multiplier is between 3 and 10. Since Fig. 7 is already quite full, adding more lines is probably not a good idea.

Conclusion: It might be important to mention that the RS multiplier is somewhat artificial and does not directly reflect observations of this process and is primarily used to achieve

the observed ice particle concentrations. This point is particularly relevant regarding the study of Seidel et al. (2024), which questions the RS efficiency.

We clarified that the multiplier is artificial.

Lines 393–394: Yes, other SIP processes may occur at colder temperatures, but it is important to keep in mind that collisional breakup can also happen between -3 and -8°C like RS or even between 0 and –3°C, unlike RS.

We tested the other SIP processes and explain the findings in the revised manuscript.

**Minor comments:**

Lines 87-90: The first sentence indicates that DS may occur, whereas the second one suggests it does not. I understand the message but this is a bit is confusing and could be clarified.

This text is now deleted as DS is included in the revised SALSA simulations.

Lines 196-200: could be moved to the previous section about SALSA microphysics.

Done.

Lines 383-386: The Lawson et al. (2015) parametrization has no temperature dependency, it is only parameterized as as function of drop size. But yes the DS process is fore sure dependent on temperature as showed for example recent experiments of Keinert et al. (2020). (https://doi.org/10.1175/JAS-D-20-0081.1)

We modified the discussion as DS (not based on Lawson et al., 2015) is actually tested and found irrelevant.

Lines 390-391: In my opinion, it is more due to the fact that no parametrization was available for DS until the paper of Lawson et al. (2015) and that the RS parametrization was used and available since decades.

We modified the discussion as DS (not based on Lawson et al., 2015) is actually tested and found irrelevant.